Prediction of antiviral drugs against African swine fever viruses based on protein–protein interaction analysis

Zhu Zhaozhong 1
Fan Yunshi 1
Liu Yang 2
Jiang Taijiao 3
Cao Yang 2 cao@scu.edu.cn
Peng Yousong 1 pys2013@hnu.edu.cn
1 College of Biology, Hunan Provincial Key Laboratory of Medical Virology, Bioinformatics Center, Hunan University , Changsha , China
2 Center of Growth, Metabolism and Aging, Key Laboratory of Bio-Resource and Eco-Environment of Ministry of Education, College of Life Sciences, Sichuan University , Chengdu , China
3 Center for Systems Medicine, Institute of Basic Medical Sciences, Suzhou Institute of Systems Medicine, Chinese Academy of Medical Sciences & Peking Union Medical College , Suzhou , China
de Azevedo Walter Jr
Electronic publication date: 2020 Apr 1
Publication date: 2020
Volume: 8
Electronic Location ID: e8855
Received 2019 Nov 19; Accepted 2020 Mar 5
Copyright: © 2020 Zhu et al.
Copyright year: 2020
Copyright holder: Zhu et al.
License: This is an open access article distributed under the terms of the Creative Commons Attribution License, which permits unrestricted use, distribution, reproduction and adaptation in any medium and for any purpose provided that it is properly attributed. For attribution, the original author(s), title, publication source (PeerJ) and either DOI or URL of the article must be cited.
License URL: https://creativecommons.org/licenses/by/4.0/

Keywords: Drug, ASFV, Prediction, PPI, Network, Interaction

Funding: National Key Plan for Scientific Research and Development of China 2016YFD0500300 Hunan Provincial Natural Science Foundation of China 2018JJ3039 National Natural Science Foundation of China 31500126 and 31671371 Chinese Academy of Medical Sciences 2016-I2M-1-005 African Swine Fever 2018NZ0151 This work was supported by the National Key Plan for Scientific Research and Development of China (2016YFD0500300), the Hunan Provincial Natural Science Foundation of China (2018JJ3039), the National Natural Science Foundation of China (31500126 and 31671371), the Chinese Academy of Medical Sciences (2016-I2M-1-005), and the funding for prevention and control technology of African swine fever (2018NZ0151). The funders had no role in study design, data collection and analysis, decision to publish, or preparation of the manuscript.

==============================
The African swine fever virus (ASFV) has severely influenced the swine industry of the world. Unfortunately, there is currently no effective antiviral drug or vaccine against the virus. Identification of new anti-ASFV drugs is urgently needed. Here, an up-to-date set of protein–protein interactions between ASFV and swine were curated by integration of protein–protein interactions from multiple sources. Thirty-eight swine proteins were observed to interact with ASFVs and were defined as ASFV-interacting swine proteins. The ASFV-interacting swine proteins were found to play a central role in the swine protein–protein interaction network, with significant larger degree, betweenness and smaller shortest path length than other swine proteins. Some of ASFV-interacting swine proteins also interacted with several other viruses and could be taken as potential targets of drugs for broad-spectrum effect, such as HSP90AB1. Finally, the antiviral drugs which targeted ASFV-interacting swine proteins and ASFV proteins were predicted. Several drugs with either broad-spectrum effect or high specificity on ASFV-interacting swine proteins were identified, such as Polaprezinc and Geldanamycin. Structural modeling and molecular dynamics simulation showed that Geldanamycin could bind with swine HSP90AB1 stably. This work could not only deepen our understanding towards the ASFV-swine interactions, but also help for the development of effective antiviral drugs against the ASFVs.

Introduction

African swine fever virus (ASFV), the causative agent of African swine fever (ASF), is an enveloped and double-stranded DNA virus with genome size ranging from 170 kbp to 194 kbp (Alonso et al., 2018). ASFV mainly infect suids and soft ticks (Sánchez-Cordón et al., 2018). In swine populations, the virus can cause 100% mortality and severely influence the swine industry (Revilla, Perez-Nunez & Richt, 2018). The ASFV has caused ASF outbreaks in more than 50 countries in Africa, Europe, Asia and South America until now (Costard et al., 2013; Malogolovkin et al., 2012). The latest reports showed that the virus has caused outbreaks in all provinces of mainland China (Zhou et al., 2018). How to effectively control the virus is still a great challenge for the globe (Wang et al., 2018).

Vaccine and antiviral drugs are believed to be the best tool for prevention and control of viral infection and spread (Monto, 2006). Unfortunately, all the attempts to develop effective vaccines against ASFVs had failed (Sánchez, Pérez-Núñez & Revilla, 2019). In the absence of vaccines against ASFVs, the antiviral drugs could not only improve host survival, but also help control the epidemic area (Zakaryan & Revilla, 2016). Therefore, it is in great need to develop effective antiviral drugs against ASFVs. Several studies have identified multiple compounds which could inhibit ASFV infections. They could be classified into two groups (Arabyan et al., 2019). The first group of antiviral drugs has identified targets and known mechanisms, and includes five kinds of drugs: (i) nucleoside analogs, such as iododeoxyuridine (Gil-Fernández et al., 1979), (S)-HPMPA (Gil-Fernández et al., 1987) and Rigid amphipathic fusion inhibitors (Hakobyan et al., 2018); (ii) interferons (IFNs) and small peptides, such as IFN-alpha (Paez, Garcia & Fernandez, 1990), IFN-gamma (Esparza, González & Viñuela, 1988) and small peptide inhibitors which could disrupt the interaction between cytoplasmic dynein and viral p54 protein (Hernáez et al., 2010); (iii) plant-derived compounds, such as genistein (Arabyan et al., 2018) and genkwanin (Hakobyan et al., 2019). The former interferes with viral type II topoisomerase and the latter disrupts the virus movement along microtubules; (iv) antibiotics, such as Rifampicin (Dardiri, Bachrach & Heller, 1971) and fluoroquinolone (Freitas et al., 2016; Mottola et al., 2013). They inhibit viral DNA-dependent RNA polymerase and type II topoisomerases, respectively; (v) small interfering RNA and CRISPR/Cas9. For example, some siRNAs (Keita, Heath & Albina, 2010) which targeted viral genes such as A151R and B646L could significantly reduce the virus titer and RNA transcripts. The other group of antiviral drugs have unknown targets and unknown mechanism, such as Apigenin (Hakobyan et al., 2016), resveratrol (Galindo et al., 2011) and oxyresveratrol (Galindo et al., 2011). However, all the antiviral drugs mentioned above have not been taken forward for commercial production. More candidate drugs are needed for further development.

Although most antiviral drugs target the viral proteins, in recent years several studies have attempted to develop antiviral drugs which targeted the host proteins (Mak et al., 2019; Yang et al., 2019). Compared to the drugs which target viral proteins, the drugs targeting host proteins have more targets in the host cell since the number of host proteins is much larger than that of viral proteins. Besides, they may be more mutant-insensitive since the host proteins evolve much slower than viral proteins (Luo, Vasudevan & Lescar, 2015; Rozenblatt-Rosen et al., 2012). With the rapid development of high-throughput assays, a large amount of protein–protein interactions between virus and host has been accumulated. Analysis of these protein–protein interactions in the perspective of network can help identify host proteins of importance for viral infection, which could be taken as potential targets for antiviral research (Uetz et al., 2006). For example, Han et al. (2017) predicted several antiviral drugs against human enterovirus 71 by systematic identification and analysis of protein–protein interactions between the virus and the host, suggesting the important role of protein–protein interaction analysis on developing antiviral drugs targeting host proteins.

Several studies have investigated the protein–protein interactions between ASFV and swine. Some important interactions were listed as follows: the viral DP71L protein interacts specifically with and activates protein phosphatase 1 (PP1) (Rivera et al., 2007); the viral A224L interacts with the proteolytic fragment of caspase-3 and inhibits the activity of this protease during ASFV infection (Nogal et al., 2001); the viral p54 binds to the light chain of cytoplasmic dynein (LC8) to hijack the microtubule motor complex during ASFV infection (Alonso et al., 2001); the viral A238L binds to the catalytic subunit of calcineurin and inhibits NFAT-regulated gene transcription in vivo (Miskin et al., 1998); the viral p30 may down-regulate the mRNA translation of host cells by interacting with hnRNP-K (Hernaez, Escribano & Alonso, 2008); the viral Ep152R interacts with BAG6 to block the immune response during viral infection (Borca et al., 2016); the viral A179L protein was able to interact with the main core Bcl-2 proapoptotic proteins Bax and Bak, and may play an important role during productive ASFV infection (Brun et al., 1996; Galindo et al., 2008). This study firstly curated a set of protein–protein interactions between ASFV and swine proteins by integration of protein–protein interactions from public databases and literatures; then, the swine proteins related to ASFV infection were identified; their roles in swine protein–protein interaction network and in interacting with other viruses, and their functions were further investigated; finally, the candidate antiviral drugs targeting these swine proteins and ASFV proteins were predicted. This work could not only deepen our understanding towards the ASFV-swine interactions, but also help for the development of effective antiviral drugs against the ASFVs.

Materials and Methods

Protein–protein interactions between ASFV and swine proteins

The protein–protein interactions between ASFV and swine proteins were compiled from three sources (Table S1). First of all, 24 protein–protein interactions with median confidence (scores greater than 0.4) between ASFV and swine, were obtained from the database of Viruses.STRING (Cook et al., 2018) on 8 January 2019.

Secondly, 20 protein–protein interactions between ASFV and swine were obtained from the literature. This was achieved by firstly searching the PubMed database by the key word “ASFV” in the title or abstract on 29 December 2018, which resulted in 630 abstracts. Then, each abstract was manually screened based on whether it contained protein–protein interactions between ASFV and swine, and 117 abstracts were retained. Finally, the full texts of the manuscripts corresponding to these abstracts were read carefully and 20 extra protein–protein interactions between ASFV and swine were compiled from these articles.

Thirdly, three protein–protein interactions between ASFV and swine proteins were inferred based on sequence homology. This was conducted by firstly collecting viral proteins (except the ASFV) which interacted with swine proteins based on the database of Viruses.STRING. Then, 159 ASFV proteins encoded by BA71V, which were downloaded from NCBI RefSeq database (Pruitt, Tatusova & Maglott, 2005), were blast (Altschul et al., 1990) against these viral proteins. The hits with e-value smaller than 0.001, coverage greater than 40%, and sequence identity greater than 30%, were considered as homologs of ASFV proteins. The swine proteins which interacted with the hits were predicted to interact with the ASFV proteins. The obtained three protein–protein interactions overlapped with those from Viruses.STRING (Table S1).

Swine protein–protein interaction network

All the swine protein–protein interactions were downloaded from STRING database (Szklarczyk et al., 2016) on 8 January 2019. Only the protein–protein interactions with a median confidence (score greater than 0.4) were kept. Besides, the redundant protein–protein interactions were removed. Finally, a protein–protein interaction network which consisted of 731,174 non-redundant protein–protein interactions between 18,683 swine proteins was obtained for further analysis.

Network analysis and visualization

The igraph package (version 1.2.2) (Csardi & Nepusz, 2006) in R was used to analyze the topology of the protein–protein interaction network. The degree and betweenness of proteins in the protein–protein interaction network were calculated with the functions of degree() and betweenness(), respectively. The shortest path length between two proteins in the protein–protein interaction network was calculated with the function of shortest.paths().

The network was visualized with the help of Cytoscape (version 3.7.1) (Su et al., 2014).

Functional enrichment analysis

The Gene Ontology (GO) terms and KEGG pathway enrichment analysis for the ASFV-interacting swine proteins or the ASFV infection-associated swine proteins were conducted with functions of enrichGO() and enrichKEGG() in the package “clusterProfiler” (version 3.6.0) (Yu et al., 2012) in R (version 3.4.2). All the GO terms and KEGG pathways with adjusted p-values smaller than 0.01 were considered as significant enrichment (Table S2).

Prediction of candidate drugs targeting ASFV and swine proteins

Candidate drugs were predicted with the help of DrugBank (version 5.1.2) (Wishart et al., 2018). The protein sequence of each ASFV protein encoded by BA71V, and that of each ASFV-Interacting swine Protein was queried against DrugBank for similar targets with the default parameters. The drugs targeting the best hit were considered to be candidate drugs for the query protein. The properties of drugs, such as the type and group of drug and ATC code, were also obtained from DrugBank (Table S3).

Modeling the interaction between Geldanamycin and swine HSP90AB1

The 3D structure of swine HSP90AB1 was predicted using homology modeling method (Eswar et al., 2006). The modeling template was the crystal structure of the Geldanamycin-binding domain of a heat shock protein 90-alpha (HSP90AA1) from Homo sapiens (PDB code: 1YET). Sequence alignment showed that the identity between swine HSP90AB1 and human HSP90AA1 was 92.3% in the Geldanamycin-binding domain (208 residues). Besides, only amino acid substitutions but no gaps were observed in the alignment. The highly similar and gap-free alignment indicated that the predicted structure is reliable. In addition, 1YET is the complex structure of Geldanamycin and HSP90AA1, which allowed us to transfer the binding conformation of Geldanamycin from 1YET to the predicted structure of swine HSP90AB1. To validate the binding conformation between Geldanamycin and swine HSP90AB1, molecular dynamics (MD) simulation was performed for 10 ns using GROMACS (Abraham et al., 2015). The RMSDs (root mean square deviation) and binding energies of the complex between Geldanamycin and swine HSP90AB1 were calculated.

Results

Interactions between ASFV and swine proteins

We firstly attempted to collect the interactions between ASFV and swine proteins as more as possible. In total, we obtained 44 protein–protein interactions between them (Fig. 1A), including 24 protein–protein interactions from the database of Viruses.STRING, 20 protein–protein interactions from the literature and three protein–protein interactions inferred from protein to protein interactions between other viruses and swine based on sequence homology (details in “Materials and Methods”). A total of 16 ASFV proteins were involved in the protein–protein interactions. Half of ASFV proteins interacted with only one swine protein. For the remaining half of ASFV proteins, the DNA-directed DNA polymerase interacted with 13 swine proteins, while the A179L and A238L both interacted with four swine proteins. Thirty-eight swine proteins were involved in the protein–protein interactions between ASFV and swine, which were defined as ASFV-interacting swine proteins. All of them only interacted with one ASFV protein except the proteins of DNAJA3, FBXO2 and SNAPIN.

Figure 1 Overview of protein–protein interactions between the ASFV and swine.

(A) Collected protein–protein interactions between ASFV and swine proteins. AIP, ASFV-interacting swine proteins. (B) All the ASFV proteins involved in protein–protein interactions and the number of interacted swine proteins. PPI, protein–protein interaction.

Construction of the ASFV-swine protein interaction network and topological analysis

To investigate the role of ASFV-interacting swine proteins in the swine, a swine protein–protein interaction network was constructed from the STRING database, which contained 731,174 non-redundant protein–protein interactions between 18,683 swine proteins. 35 (94%) ASFV-interacting swine proteins were found to interact with 4,184 other swine proteins. The latter 4,184 swine proteins were defined as ASFV infection-associated swine proteins. The ASFV-interacting swine proteins together with ASFV infection-associated swine proteins formed a protein interaction network with 9,305 non-redundant interactions (Fig. 2A), including 68 interactions between ASFV-interacting swine proteins.

Figure 2 The protein–protein interaction network between ASFV-interacting swine proteins and ASFV infection-associated proteins, and the topological analysis of these proteins.

(A) The protein–protein interaction network between ASFV-interacting swine proteins (AIPs) and ASFV infection-associated proteins (AAPs). (B–D) Distribution of the betweenness centrality, degree and shortest path length for all proteins (ALL), ASFV-interacting swine proteins (AIPs) and ASFV infection-associated swine proteins (AAPs) in the swine protein–protein interaction network.

To investigate the centrality of ASFV-interacting and ASFV infection-associated swine proteins in the swine protein–protein interaction network, we calculated the degree and betweenness centrality, and the average shortest path length of each protein in the swine protein–protein interaction network (Figs. 2B–2D). The node (protein) degree was defined as the number of connections the node has to other nodes in the network; the node betweenness was defined as the number of shortest paths that pass through the node; the average shortest path length of a node was defined as the average length of all shortest paths from the node to other nodes. All of them measure the importance of a node in the network. The median degree and betweenness centrality, and the median shortest path length of all proteins in the swine protein–protein interaction network were 32, 103.5 and 59.2, respectively, whereas these values for the ASFV-interacting swine proteins were 144, 104.6 and 58.7, respectively, and they were 120, 104.2 and 58.8, respectively, for the ASFV infection-associated swine proteins (Fig. 2B). The ASFV-interacting swine proteins and ASFV infection-associated swine proteins were observed to have significant larger degrees and betweenness, and smaller shortest path length than all swine proteins, with p-values much smaller than 0.001 in the two-sided Wilcoxon rank-sum test. This suggested that the ASFV-interacting swine proteins and ASFV infection-associated swine proteins played a central role in the swine protein–protein interaction network.

Functional analysis of ASFV-interacting swine proteins and ASFV infection-associated swine proteins

Since the ASFV-interacting swine proteins and ASFV infection-associated swine proteins were observed to play a central role in swine protein–protein interaction network, we next investigated their functions. Functional enrichment analysis was conducted on the ASFV-interacting swine proteins and ASFV infection-associated swine proteins (Table S2). Only a few GO terms in the domain of Molecular Function were enriched. Interestingly, nearly 50 KEGG pathways were enriched in the ASFV-interacting swine proteins (Table S2). The pathways of Necroptosis and Alcoholism were two of the most enriched pathways, both of which included more than 20% of all ASFV-interacting swine proteins. Besides, three pathways related to virus infection, such as “Herpes simplex virus 1 infection”, were also enriched.

We further conducted the functional enrichment analysis on the ASFV infection-associated swine proteins. Figures 3A–3D showed the top ten GO terms in three domains of GO and KEGG pathways enriched in the ASFV infection-associated swine proteins. In the domain of Biological Process, six of top 10 enriched GO terms were related to cell death or apoptotic process; in the domain of Cellular Component, the ASFV infection-associated swine proteins were enriched in the nuclear and cytoskeleton; in the domain of Molecular Function, the ASFV infection-associated swine proteins were enriched in the GO terms of binding and enzyme activity. For the KEGG pathways, “Herpes simplex virus 1 infection” was most enriched. Besides, several signaling pathways were also enriched, such as “PI3K−Akt signaling pathway” and “MAPK signaling pathway”.

Figure 3 Functional enrichment analysis of AAPs.

(A–D) Top 10 enriched terms in the domain of biological process, cellular component and molecular function and KEGG pathways were shown.

The role of ASFV-interacting swine proteins in the protein–protein interactions between swine and other viruses

We then investigated the role of ASFV-interacting swine proteins in the protein–protein interactions between swine and other viruses. All the protein–protein interactions between ASFV-interacting swine proteins and other viral proteins which were public available in the Viruses.STRING database were obtained. As was shown in Fig. 4A, 48 protein–protein interactions were obtained and shaped a network, which included 15 proteins from 11 other viruses (nodes in square) and 16 ASFV-interacting swine proteins (nodes in ellipse). A total of 13 ASFV-interacting swine proteins were observed to interact with more than one other virus. Besides, they also interacted with several hundreds of proteins in the swine protein–protein interaction network (Fig. 4B). For example, the heat shock protein 90 s, including HSP90AB1, HSP90AA1 and HSP90B1, could interact with proteins from other five viruses, and interact with more than 1,500 swine proteins, suggesting their central roles in both the virus-swine protein–protein interaction network and swine protein–protein interaction network.

Figure 4 ASFV-interacting swine proteins and their interactions with other viruses.

(A) The protein–protein interaction network between ASFV-interacting swine proteins (AIPs) and other viruses. ASFV-interacting swine proteins were represented as ellipse in gray. Viruses were represented as squares and colored according to the legend in the bottom right. VESE, Vesicular exanthema of swine virus; Swinepox, Swinepox virus; TTSV1a, Torque teno sus virus 1a; TeschoA, Teschovirus A; Nodamura, Nodamura virus; FMDV, Foot-and-mouth disease virus; TTSVk2, Torque teno sus virus k2; FLUCV, Influenza C virus; EMCV, Encephalomyocarditis virus. (B) The number of interacted virus and the degree of ASFV-interacting swine proteins in the swine protein–protein interaction network. HISTH2AC, histone H2A type 2-C; H2A1-H, histone H2A type 1-H; HIST1H2AA, histone H2A type 1-A; HIST1H2AJ, histone H2A type 1-like; H2A2-A-L, histone H2A type 2-A-like; H2AFX, histone H2AX.

Drug prediction for treating ASFVs

The wide involvement of ASFV-interacting swine proteins in protein–protein interactions between swine and multiple viruses, and the central role of ASFV-interacting swine proteins in swine protein–protein interaction network, suggested the possibility of their use as broad-spectrum host-dependent antiviral drug targets. Therefore, we attempted to predict drugs targeting the ASFV-interacting swine proteins with the help of DrugBank (Table S3). As was shown in Fig. 5A total of 142 drugs (in ellipse or square) were predicted to target 21 ASFV-interacting swine proteins (pink circles). Most of the drugs were small molecules (colored ellipses); the other drugs were protein or peptide (colored squares). Some ASFV-interacting swine proteins were targeted by multiple drugs, such as the heat shock protein 90 alpha family class A member 1 (HSP90AA1) and tumor necrosis factor (TNF). HSP90AA1 was targeted by more than 30 drugs, most of which were small molecules and were in experimental; while TNF were also targeted by more than 30 drugs, most of which were approved or investigational.

Figure 5 Predicted drugs targeting the ASFV-interacting swine proteins (AIPs) and ASFV proteins.

The interactions above and below the dotted line referred to those between drugs and ASFV-interacting swine proteins, and those between drugs and ASFV proteins, respectively. The ASFV-interacting swine proteins and ASFV proteins were represented as red and cyan circles, respectively. Drugs of protein or peptide, and those of small molecule, were represented as squares and ellipses, respectively. Drugs in the stage of approved, investigational and experimental were colored in orange, purple and light green, respectively. Drugs which specifically targeted one ASFV-interacting swine protein were highlighted in black-edge. Two drugs which targeted both the ASFV-interacting swine protein and ASFV protein were highlighted in red-edge.

We also predicted drugs targeting the ASFV proteins. Twenty-nine small molecules were predicted to target 10 ASFV proteins. Among them, both the proteins of F778R and A240L were targeted by eight drugs. However, these ten ASFV proteins were not involved in the protein–protein interactions between ASFV and swine. Interestingly, the Gallium nitrate (DrugBank ID: DB05260), a drug used for treating hyper-calcemia and Rifabutin (DrugBank ID: DB00615), a antibiotic with potent antimycobacterial properties, were observed to both target the ASFV-interacting swine proteins and ASFV proteins. Both of them (highlighted in red-edge) were approved for use, suggesting their potential use for treating the ASFVs.

Some drugs were observed to have strong specificity on the ASFV protein or ASFV-interacting swine proteins, such as the Hydroxyurea (DB01005), Infliximab (DB00065), Adalimumab (DB00051) and so on. Hydroxyurea specifically targeted F778R. It is an antineoplastic agent that inhibits DNA synthesis through the inhibition of ribonucleoside diphosphate reductase. It may be used to inhibit DNA synthesis of the ASFV virus, thus blocking the proliferation of the virus. Infliximab specifically targeted TNF and is primarily related to inflammation control and neurological indications. It may be used to block the necrosis during the ASFV infection.

Some drugs were observed to target multiple ASFV-interacting swine proteins, such as Geldanamycin (DB02424), Polaprezinc (DB09221) and Andrographolide (DB05767). For example, Geldanamycin could target the HSP90AA1, HSP90AB1 and HSP90B1, all of which played a central role in the swine protein–protein interaction network and swine-virus protein–protein interaction network (Fig. 4B). This suggested that Geldanamycin may have a broad-spectrum effect in disrupting the swine-ASFV protein–protein interaction network, and may inhibit the viral infections effectively. Therefore, we further investigated the mechanism of Geldanamycin by modeling the interactions between Geldanamycin and swine HSP90AB1 (Fig. 6). The 3D structure of the Geldanamycin-binding domain of swine HSP90AB1 was modeled using the template of human HSP90AA1 (PDB code: 1YET). The binding conformation of Geldanamycin was also predicted by referring the same compound in the template structure (Fig. 6A) (see “Methods”). As the sequence identity between the template and swine HSP90AB1 was as high as 92.8%, particularly in the ligand-binding regions, their binding modes with Geldanamycin in two structures should be highly similar. MD was performed for the complex structure of Geldanamycin and swine HSP90AB1 to investigate their interactions. The result showed that the RMSD (root mean square deviation) of Geldanamycin and swine HSP90AB1 were less than 1.6 Å in the 10 ns simulation (Fig. 6B), which indicated the binding conformation was highly stable. The key contributions of binding energy were from ASN51, ASP54, LYS58, ASP93, MET98, ASP102, ASN106, PHE138 and THR184, which was almost identical to the template (Fig. S1). In addition, the binding free energy for swine HSP90AB1 and Geldanamycin complex structures was −76.8 kcal/mol which was even stronger than −69.1 kcal/mol of the template (Fig. S2). Hence, Geldanamycin could bind with swine HSP90AB1, similar to that in Homo sapiens.

Figure 6 Modelling the interactions between Geldanamycin and swine HSP90AB1.

(A) The predicted 3D structure of the Geldanamycin-binding domain of swine HSP90AB1 and its interaction with Geldanamycin. (B) RMSD of all Cα atoms for the ligand (black) and receptor-ligand complex (red) during MD simulations (10 ns).

Discussion

Vaccines and antiviral-drugs are considered as the most effective tools for fighting against viruses. Unfortunately, nearly all attempts to develop vaccines against ASFVs have failed to induce effective protection (Wang et al., 2018). Therefore, it is necessary to develop antiviral drugs against the virus. Previous studies have found several antiviral drugs which could possibly inhibit ASFV infection in vitro (Arabyan et al., 2019), including nucleoside analogs (Berry & Kinsella, 2001), genistein (Arabyan et al., 2018), genkwanin (Hakobyan et al., 2019), Rifampicin (Dardiri, Bachrach & Heller, 1971), fluoroquinolone (Mottola et al., 2013), sulfated polysaccharides (García-Villalón & Gil-Fernández, 1991), lauryl gallate (Hurtado et al., 2008), stilbenes resveratrol (Galindo et al., 2011), oxyresveratrol (Galindo et al., 2011), histone deacetylases enzymes (HDACs) inhibitor (Frouco et al., 2017), small peptide inhibitors (Hernáez et al., 2010) and so on. This study computationally predicted several candidate drugs targeting the ASFV proteins and ASFV-interacting swine proteins, which may be helpful for the development of more effective drugs against the ASFV.

In the era of systems biology, a large amount of protein–protein interactions have been accumulated, including the virus-host protein–protein interactions. This study compiled an up-to-date protein–protein interaction network between ASFV and swine. Analysis of the network could help identify possible associations between viral activities and host defense strategies, which may facilitate development of potential therapies by disrupting host-virus interactions (Tan et al., 2007). Several ASFV-interacting swine proteins and lots of ASFV infection-associated swine proteins were identified based on the protein–protein interaction network. They were observed to interact with more proteins or have larger influences on the information flow throughout the swine protein–protein interaction network than other proteins, suggesting their central roles in the swine protein–protein interaction network. Some ASFV-interacting swine proteins were observed to interact with multiple viruses. They could be used as antiviral drug targets. Besides, the predicted drugs targeting these ASFV-interacting swine proteins, such as Polaprezinc and Geldanamycin, may have a broad-spectrum effect against viral infections.

Both the ASFV-interacting swine proteins and ASFV infection-associated swine proteins were enriched in the functions of cell death, apoptosis or necroptosis. This suggested that these processes may play an important role in viral infections. Previous studies have shown that ASFV infection induced TNF-alpha production which further induced apoptosis in the infected cell (Del Moral et al., 1999). Actually, ASFV induced apoptosis of infected cells both in vitro and in vivo (Ramiro-Ibáñez et al., 1996). ASFV has several strategies to regulate apoptosis. It encodes several anti-apoptotic proteins such as A179L and A224L to delay the execution step of the apoptotic pathway. The virus also has the capacity of regulating the unfolded protein response to prevent early apoptosis and ensure viral replication (Galindo et al., 2012). In the late stage of ASFV infection, induction of apoptosis could favor virus spread without the activation of inflammatory responses (Carrascosa et al., 2002). Therefore, the drugs which could inhibit cell death, apoptosis or necroptosis, may be candidates for treatment of ASFVs. For example, the infliximab, which is a TNF blocker and primarily related to inflammation control (Keane et al., 2001), may be used to block the necrosis during the ASFV infection.

Lots of drugs were predicted to target the ASFV-interacting swine proteins or ASFV proteins. Several strategies could be used to select the candidate drugs. For specificity, the drugs with high specificity on the ASFV-interacting swine proteins or ASFV proteins could be selected, such as the Hydroxyurea and Infliximab; for broad-spectrum effect, the drugs which targeted the ASFV-interacting swine proteins with high degrees in the swine protein–protein interaction network, such as Polaprezinc, or those which targeted multiple ASFV-interacting swine proteins, such as Geldanamycin, could have a large influence on the protein–protein interactions between swine and ASFV. Two drugs, that is, Gallium nitrate and Rifabutin, were observed to target both the ASFV-interacting swine proteins and ASFV proteins. They could also be used for broad-spectrum inhibitory effect against ASFV infections.

Most antiviral drugs target the viral proteins. Drug resistance frequently appears due to rapid mutation of viruses. On the contrary, the drugs targeting the host protein may have the advantage of stable effect since the host proteins generally evolve far slower than viral proteins (He, Duan & Tan, 2007; Tavassoli, 2011). Besides, some host proteins may interact with multiple viruses, such as HSP90AB1 mentioned above. The drugs targeting them may have broad-spectrum antiviral effect. Bioinformatics analysis of the accumulated protein–protein interactions between virus and host cell can facilitate the identification host proteins which are vital for viral infection. As the accumulation of protein–protein interactions and the rapid development of bioinformatics methods, several antiviral molecules with reduced side effects have been proposed and validated (Sessions et al., 2009). Previous studies have developed antiviral drugs against the host proteins such as the ubiquitin-proteasome system (Barrado-Gil et al., 2017) and HDACs (Frouco et al., 2017) to inhibit the replication of ASFVs. This study investigated the prediction of antiviral drugs against ASFV infections. The ASFV-interacting swine proteins could be taken as potential antiviral-drug targets. The candidate drugs identified here may facilitate further development of effective drugs against the virus.

There were two limitations to this study. Firstly, the protein–protein interactions between swine and ASFV are far from complete. The ASFV encodes more than 150 proteins (Rodríguez & Salas, 2013). Only 16 of them were involved in the protein–protein interactions analyzed here. Much more efforts are needed to generate a comprehensive protein–protein interaction network between swine and ASFV. Fortunately, based on the limited protein–protein interactions between swine and ASFV, several antiviral drugs were predicted and had the potential for further development. Secondly, the drugs predicted here need further experimental validations. Several drugs with high specificity on ASFV-interacting swine proteins, or with broad-spectrum effect, such as Polaprezinc and Geldanamycin, could be prioritized for validation.

Conclusions

In conclusion, this study curated an up-to-date set of protein–protein interactions between swine and ASFV as far as we know, and identified the ASFV-interacting swine proteins which were vital for viral infection. The ASFV-interacting swine proteins were observed to play a central role in swine protein–protein interaction network, and also took part in interactions between swine and several other viruses. They could be taken as potential antiviral-drug targets. Several drugs were predicted to target the ASFV-interacting swine proteins and ASFV proteins. They could be helpful for further development of effective drugs against the virus.

Supplemental Information

Supplemental Information 1 Contribution energy for each residue in the template (A) and modeled complex between Geldanamycin and swine HSP90AB1 (B).

MM: contains Van der Waals, electrostatic interactions, and net non-bonded potential energy between the protein and inhibitor; Polar: polar solvation energy; Apolar: non-polar solvation energy; Total: total energy.

Click here for additional data file.

Supplemental Information 2 Binding energy of receptor-ligand complex for the template (A) and modeled complex between Geldanamycin and swine HSP90AB1 (B) during MD simulations (10 ns).

ΔEvdw refer to Van der Waal energy, ΔEelec refer to electrostatic energy, ΔEmm refer to total potential energy between the protein and ligand (ΔEmm = ΔEvdw + ΔEelec), ΔEpol refer to polar solvation energy, ΔEapol refer to non-polar solvation energy and ΔEbinding refer to the total binding energy (ΔEbinding = ΔEmm + ΔEpol + ΔEapol).

Click here for additional data file.

Supplemental Information 3 The curated PPIs between ASFV and swine.

Click here for additional data file.

Supplemental Information 4 The enriched GO terms and KEGG pathways for AIPs and AAPs.

Click here for additional data file.

Supplemental Information 5 The predicted drugs targeting the AIPs and ASFV proteins.

Click here for additional data file.

Supplemental Information 6 Raw data for Figure 6.

Click here for additional data file.

Additional Information and Declarations

Competing Interests

Author Contributions

Data Availability

The authors declare that they have no competing interests.

Zhaozhong Zhu performed the experiments, analyzed the data, prepared figures and/or tables, authored or reviewed drafts of the paper, and approved the final draft.

Yunshi Fan performed the experiments, prepared figures and/or tables, and approved the final draft.

Yang Liu analyzed the data, prepared figures and/or tables, and approved the final draft.

Taijiao Jiang conceived and designed the experiments, authored or reviewed drafts of the paper, and approved the final draft.

Yang Cao performed the experiments, prepared figures and/or tables, and approved the final draft.

Yousong Peng conceived and designed the experiments, analyzed the data, prepared figures and/or tables, authored or reviewed drafts of the paper, and approved the final draft.

The following information was supplied regarding data availability:

The raw measurements are available in the Supplemental Files.

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
