# Peer review of "Prediction of antiviral drugs against African swine fever viruses based on protein–protein interaction analysis"

_PeerJ, doi:10.7717/peerj.8855_

## Round 0.1 · original submission · Major Revisions

Two specialists in the field evaluated your manuscript. They have several concerns related to your submission. In my view, your paper needs a major revision.

Reviewer 1 ·

Basic reporting

In abstract please define AIPs;

In introduction please:
1) add additional references presenting commercial drugs with anti-ASFV proprieties (e.g. 10.1016/j.antiviral.2016.08.021; 10.1016/j.vetmic.2013.01.018) and also in the discussion section (anti-topoisomerase poisons;
2) clarify the possible role of antivirals in the ASFV context (including outbreaks);
3) line 61: describe "multiple sources";

Also in discussion, references of drugs acting against cell proteins (e.g. ubiquitin-proteasome system, HDACs), with anti-ASFV-proprieties should be citated (e.g. 10.1371/journal.pone.0189741; 10.1016/j.virusres.2017.09.009)

Experimental design

The manuscript is in the scope of the Journal. Research question is well defined and relevant. Although, in M&M section please:
1) justify the used theresolds levels for e-value;

Validity of the findings

To support the data, it will be very interesting if the authors could perform some in vitro assays with Geldanamycin using swine cells

Reviewer 2 ·

Basic reporting

I find that these results are interesting but authors should take the effort to discuss them further in the context of the previous literature, that is very specific for those interactions found. With some more work, the manuscript will improve significantly. A clear professional English is used in this manuscript. However, in the text, abbreviations that do not belong to the common usage for scientific readers such as PPIs, AAPs or AIPs should be avoided. These could be used in Figures instead.
Your Introduction lacks citation of important works in antivirals against ASFV I suggest that you improve your review of the current literature and at least include the following references: Galindo et al. doi: 10.1016/j.antiviral.2011.04.013 and Hernaez et al. doi: 10.1128/JVI.01168-10

In Intro section, you should substitute Citation #1 and #9. #1 should be better cited with a publication related with ASFV virology such as ICTV Virus Taxonomy Profile: Asfarviridae Journal of General Virology doi 10.1099/jgv.0.001049
Overall the manuscript, authors rather reference publications related to other viruses instead ASFV.
Figures are relevant and high quality but could be better labeled and described in figure legends and results. A really good adding to this manuscript could be to include the references that previously described the interactions that they curated in the Figure 1.
It could be useful for readers to include a description of the main interactions and the corresponding citations:
Ep152R could be related with its reference: Borca M.V., V. O’Donnell, L.G. Holinka, D.K. Rai, B. Stanford, M. Alfano, J. Carlson, P.A. Azzinaro, C. Alonso and D.P. Gladue. 2016. The Ep152R ORF of African swine fever virus strain Georgia encodes for an essential gene that interacts with host protein BAG6. Virus Research 223: 181-189
DP71L could be related to its references: Rivera J., C Abrahams, B Hernaez, A Alcázar, L Dixon, J M. Escribano and C Alonso. 2007. The MyD116-African Swine Fever viral homologue interacts with the catalytic subunit of protein phosphatase-1 and activates its phosphatase activity. Journal of Virology 81 (6): 2923- 2929.
A179L: *Brun A., C. Rivas, M. Esteban, J.M. Escribano and C. Alonso. African swine fever virus gene A179L, a viral homologue of bcl-2, protects cells from programmed cell death. Virology 225, 227- 230, 1996. *Galindo I., B. Hernáez, G. Díaz-Gil, J.M. Escribano and C. Alonso. 2008. A179L, a viral Bcl-2 homologue, targets the core Bcl-2 apoptotic machinery and its upstream BH3 activators with selective binding restrictions for Bid and Noxa. Virology Vol 375(5): 561-572.
P54: Alonso C., J. Miskin, B. Hernáez, P. Fernandez-Zapatero, L. Soto, C. Cantó, I. Rodríguez-Crespo, L. Dixon, J.M. Escribano. 2001. The African swine fever virus protein p54 interacts with the microtubule motor complex through direct binding to light chain dynein. Journal of Virology 75 (20), 9819 -9827.
P30 Hernáez B., J M. Escribano and C Alonso. 2008. The African swine fever virus protein p30 interacts with the heterogeneous ribonucleoprotein K (hnRNP-K) during infection. FEBS Letters 582, 3275–3280, 2008.
A238L could be related with its references, which are several: Miskin JE, Abrams CC, Goatley LC, Dixon LK. A viral mechanism for inhibition of the cellular phosphatase calcineurin. Science. 1998 Jul 24;281(5376):562-5.
A224L Nogal ML, González de Buitrago G, Rodríguez C, Cubelos B, Carrascosa AL, Salas ML, Revilla Y. African swine fever virus IAP homologue inhibits caspase activation and promotes cell survival in mammalian cells. J Virol. 2001 75(6):2535-43.

Experimental design

The research is within the scope of the Journal and the research question is well very relevant and meaningful. Methods are described in detail but some more explanations for the non-expert reader would be desirable.
The results presented are interesting but lack any proof of the antiviral activity of the drugs, hence the sentence in lines 243-244 is overestimating the relevance of the results shown in this manuscript. Results require further explanations to the non-specialist reader in topological analysis.

Validity of the findings

The impact and novelty of the findings is good and relevant to the field. Data are statistically sound and controlled. Conclusions are good.

Additional comments

Your Introduction lacks citation of important works in antivirals against ASFV I suggest that you improve your review of the current literature and at least include the following references: Galindo et al. doi: 10.1016/j.antiviral.2011.04.013 and Hernaez et al. doi: 10.1128/JVI.01168-10

In Intro section, you should substitute Citation #1 and #9. #1 should be better cited with a publication related with ASFV virology such as ICTV Virus Taxonomy Profile: Asfarviridae Journal of General Virology doi 10.1099/jgv.0.001049
Overall the manuscript, authors rather reference publications related to other viruses instead ASFV.
Figures are relevant and high quality but could be better labeled and described in figure legends and results. A really good adding to this manuscript could be to include the references that previously described the interactions that they curated in the Figure 1.
It could be useful for readers to include a description of the main interactions and the corresponding citations:
Ep152R could be related with its reference: Borca M.V., V. O’Donnell, L.G. Holinka, D.K. Rai, B. Stanford, M. Alfano, J. Carlson, P.A. Azzinaro, C. Alonso and D.P. Gladue. 2016. The Ep152R ORF of African swine fever virus strain Georgia encodes for an essential gene that interacts with host protein BAG6. Virus Research 223: 181-189
DP71L could be related to its references: Rivera J., C Abrahams, B Hernaez, A Alcázar, L Dixon, J M. Escribano and C Alonso. 2007. The MyD116-African Swine Fever viral homologue interacts with the catalytic subunit of protein phosphatase-1 and activates its phosphatase activity. Journal of Virology 81 (6): 2923- 2929.
A179L: *Brun A., C. Rivas, M. Esteban, J.M. Escribano and C. Alonso. African swine fever virus gene A179L, a viral homologue of bcl-2, protects cells from programmed cell death. Virology 225, 227- 230, 1996. *Galindo I., B. Hernáez, G. Díaz-Gil, J.M. Escribano and C. Alonso. 2008. A179L, a viral Bcl-2 homologue, targets the core Bcl-2 apoptotic machinery and its upstream BH3 activators with selective binding restrictions for Bid and Noxa. Virology Vol 375(5): 561-572.
P54: Alonso C., J. Miskin, B. Hernáez, P. Fernandez-Zapatero, L. Soto, C. Cantó, I. Rodríguez-Crespo, L. Dixon, J.M. Escribano. 2001. The African swine fever virus protein p54 interacts with the microtubule motor complex through direct binding to light chain dynein. Journal of Virology 75 (20), 9819 -9827.
P30 Hernáez B., J M. Escribano and C Alonso. 2008. The African swine fever virus protein p30 interacts with the heterogeneous ribonucleoprotein K (hnRNP-K) during infection. FEBS Letters 582, 3275–3280, 2008.
A238L could be related with its references, which are several: Miskin JE, Abrams CC, Goatley LC, Dixon LK. A viral mechanism for inhibition of the cellular phosphatase calcineurin. Science. 1998 Jul 24;281(5376):562-5.
A224L Nogal ML, González de Buitrago G, Rodríguez C, Cubelos B, Carrascosa AL, Salas ML, Revilla Y. African swine fever virus IAP homologue inhibits caspase activation and promotes cell survival in mammalian cells. J Virol. 2001 75(6):2535-43.
The manuscript results interesting but lack any proof of the antiviral activity of the drugs, hence the sentence in lines 243-244 is overestimating the relevance of the results shown in this manuscript. Results require further explanations to the non-specialist reader in topological analysis.
Discussion is rather scarce and it should emphasize previous work on antivirals, given the fact that the list is not so large to resolve it adding “and so on”. Authors should search the literature for those and use the list given above.
The authors should discuss the induction of apoptosis by this virus in vivo and in vitro and include the appropriate references, not in other virus infections. However, they have included reference 30, which is clearly wrong. They should include the following: Ramiro-Ibanez et al, 1996 doi: 10.1099/0022-1317-77-9-2209 and Galindo et al, 2012 doi: 10.1038/cddis.2012.81
Also, when discussing TNF, it should be important to include the reference: Gomez del Moral M., E. Ortuño, P. Fernandez-Zapatero, F. Alonso, C. Alonso, A. Ezquerra and J. Dominguez. African swine fever virus infection induces tumor necrosis factor alpha production: Implications in pathogenesis. Journal of Virology 73: 2173-2180, 1999
Reference number 20 is wrong. In the manuscript, I have highlighted some mistakes in yellow.

Annotated reviews are not available for download in order to protect the identity of reviewers who chose to remain anonymous.

---

## Round 0.2 · accepted · Accept

The authors carried out all modifications indicated by the reviewers. In my view, the manuscript can be accepted for publication.